# The Development and Validation of the “Hippocratic Hypertension Self-Care Scale”

**DOI:** 10.3390/healthcare11182579

**Published:** 2023-09-18

**Authors:** Hero Brokalaki, Anastasia A. Chatziefstratiou, Nikolaos V. Fotos, Konstantinos Giakoumidakis, Evaggelos Chatzistamatiou

**Affiliations:** 1Department of Nursing, School of Health Sciences, National and Kapodistrian University of Athens, 11527 Athens, Greece; heropan@nurs.uoa.gr (H.B.); nikfotos@nurs.uoa.gr (N.V.F.); 2Cardiac Surgery Unit, Genral Peadiatric Hospital of Athens “Agia Sophia”, 11527 Athens, Greece; 3Department of Nursing, School of Health Sciences, Hellenic Mediterranean University, 71410 Heraklion, Greece; kongiakoumidakis@hmu.gr; 4Spiliopoulio Hospital of Athens “Saint Elena”, 11521 Athina, Greece; vagelisdoc@hotmail.com

**Keywords:** self-care, self-management, hypertension, validation, scale development

## Abstract

Background: The adoption of self-care behaviors among patients with arterial hypertension (AH) plays an important role in the management of their health condition. However, a lack of scales assessing self-care is observed. We aimed to develop and validate the Hippocratic hypertension self-care scale. Methods: From a pool of questions derived from a literature review, 18 items were included in the scale and reviewed by a committee of experts. Participants indicated the frequency at which they followed the self-behavior prescribed in each statement on a five-point Likert scale. Data were collected between April 2019 and December 2019. Results: A total of 202 consecutive adult patients with AH were enrolled in the study. The internal consistency of the scale was found to be 0.807, using Cronbach’s alpha coefficient. An exploratory factor analysis identified two domains that accounted for 92.94% of the variance in the scale items; however, each sub-scale could not be used as an independent scale. Finally, the test–retest of the scale showed a significant strong correlation (r = 0.0095, *p* < 0.001). Conclusion: This analysis indicates that the scale is reliable and valid for assessing self-care behaviors in patients with AH. It is suggested that health professionals use it in their clinical practice to improve the management of AH.

## 1. Introduction

Arterial hypertension (AH) is a chronic health condition that may be associated with the development of myocardial disease; stroke; kidney, eye, and vascular diseases. More specifically, the World Health Organization estimates that 54% of strokes and 47% of cases of ischemic heart disease are attributed to high blood pressure. It is estimated that 1.13 billion people worldwide had AH in 2015, whereas over 150 million of them were located in Central and Eastern Europe [1]. The overall prevalence of AH among adults is estimated at around 30–45%, whereas it is approximately 24% in men and 20% in women [1,2]. The prevalence is characterized by a progressive increase with advancing age since more than 60% of people over 60 years old have AH [2]. However, a significant increase in AH prevalence is expected during the next decades because of the population’s age, sedentary lifestyle, and increase in body weight. More specifically, a 15–20% rise is predicted by 2025, which corresponds to 1.5 billion people [3]. In Greece, the prevalence of AH in the general adult population is 41.7%, 45.8% in males, and 37.9% in females [4].

Studies have reported that a reduction in blood pressure (BP) can substantially decrease the total cardiovascular risk as well as all-cause mortality [5]. Also, the reduced incidence of stroke in the last decades can be accounted for, in large measure, by the decrease in blood pressure. The contribution is more significant when baseline BP levels are high. In a meta-analysis of 61 studies involving more than 1 million patients with hypertension, it was observed that a reduction in systolic and diastolic BP reduced cardiovascular events [6]. More specifically, it was found that for people between 40 and 60 years old, a reduction by 20 mmHg in systolic BP is associated with a decrease in risk for stroke and mortality of coronary heart disease. Also, the same effect of a reduction of 10 mmHg in diastolic BP was found. However, it is important to mention that a reduction in BP depends on the level at which patients follow the recommended self-behaviors, such as medication, diet, smoking, alcohol consumption, and exercise [7,8,9]. However, only a minority of patients modify their lifestyle after a diagnosis of AH, and it is hard for them to sustain changes.

Therefore, we reviewed the existing literature through electronic databases to identify the methods used to assess the level of self-behavior among patients with chronic diseases. In the next stage, we reviewed the available scales assessing self-behavior specific in patients with AH. For instance, it is observed that there are plenty of scales assessing the level of adherence to antihypertensive medication only, like the Morisky–Green scale [10], A-14 [11] scale, and Adherence to Refills and Medications Scale (ARMS) [12]. At the same time, we identified the Hill–Bone scale, which aims to evaluate not only adherence to medication but also adherence to salt consumption and appointment-keeping with healthcare providers [13]. It is important to mention that all the above scales are disease-specific for AH; however, none of them evaluates the whole aspects of self-behavior.

Therefore, the aim of the present study was to develop and assess the validity and reliability of the Hippocratic hypertension self-care scale. The goals of the study were the following:Develop the Hippocratic hypertension self-care scale;Examine the reliability of the Hippocratic hypertension self-care scale;Examine the factorial structure of the Hippocratic hypertension self-care scale;Assess the structural estimation modeling approach of the Hippocratic hypertension self-care scale with the use of explanatory factor analysis (EFA).

## 2. Materials and Methods

### 2.1. Establishment of Face and Content Validity of the Hippocratic Hypertension Self-Care Scale

Recent data from the literature and reports from international health associations like the European Society of Cardiology and the European Society of Hypertension were reviewed for the development of the scale. During the development of the Hippocratic hypertension self-care scale, an 18-item scale was prepared by the authors, which includes 5 items on medication aspects (items 1–5), 6 items on diet aspects (items 6–11), 1 item on an exercise aspect (item 12), 2 items on alcohol aspects (items 13–14), 1 item on a smoking topic (item 15), 1 item on blood pressure measurement (item 16), and 2 items on appointment keeping (items 17–18). Therefore, the scale includes 7 sub-sections. Each question was encoded in a five-point Likert scale from never (0 points) to very frequently (4 points), with the resulting total summed score ranging between 0 and 72. See Appendix A. It is important to clarify that items 1–4, 7–9, 12–15, and 17–18 were to be reverse-scored. As for the score, we used quartiles to organize data into three points—a lower quartile, median, and upper quartile—to form four groups of the dataset. More specifically, a score over 54 was classified as very good, a score between 50 and 54 as good, a score between 45 and 50 as fair, and a score below 45 as poor. Higher scores indicate that patients follow and adopt the recommended self-behaviors.

Ten items questioned how often the patients did not follow the recommended self-behaviors regarding medication, diet, and salt consumption during the last week. Six items examined how often the individuals did not follow the recommended self-behaviors concerning physical activity, alcohol consumption, body weight, smoking, and blood pressure measurement during the last month, while two items questioned how often the patients did not follow the recommended self-behaviors regarding appointment keeping during the last year.

The content validity was assessed through the evaluation of seven experts (two cardiologists, two nurses who specialized in hypertension, one expert in statistics, and two specialists in psychometrics). The professionals graded each question as “essential”, “useful but inadequate”, or “unnecessary”. All questions were assessed for clarity.

As a next step, twenty people without any research background were invited to test the scale for its language and clarity. These persons were not involved in the final sample of the study.

### 2.2. Study Population and Data Collection

The study was conducted at Hippokration General Hospital, Athens, between April 2019 and December 2019. The sample consisted of 202 men and women who visited the Hypertension Management Unit for their appointment for a routine check-up. The sample size was calculated so that the question item/participant ratio would be at least 1/10. The size of the sample was considered appropriate in order for the results of the present study to be considered adequate. Therefore, healthcare providers could be able to use the Hippocratic hypertension self-care scale without any doubt for the accuracy of their results.

The study included participants with the below criteria:

(1) Age over 18 years old;

(2) Diagnosed AH;

(3) Prescription of at least one antihypertensive drug;

(4) Able to read and write Greek;

(5) Provided written informed consent.

On the other hand, participants with the following criteria were excluded from the study:

(1) The presence of a life-threatening disease;

(2) The presence of a psychiatric disorder;

(3) A history of acute myocardial infarction during the last 2 months or cardiac surgery during the last 6 months.

During the first assessment, the study authors assembled their data via a face-to-face interview. In the second step, the researchers called the participants (*n* = 30) one month later in order for the sample to re-answer the questions (test–retest reliability). The tool was administered one month after the first assessment, so as to avoid the possibility of participants recalling their answers (memory effect) [14]. The Hippocratic hypertension self-care scale was accomplished for all participants, and demographic characteristics were evaluated. Patients needed 10 min to answer all items on the scale.

All participants enrolled in the study provided written informed consent, after receiving a complete description of the study and having the opportunity to ask for clarification. A cover letter accompanied the questionnaires, explaining the purpose of the study, providing the researchers’ affiliation and contact information, and clearly stating that the answers would be confidential and that anonymity in the final data reports was guaranteed (Ethical Committee’s approval No.: 52/21-12-2017). Participants did not receive any type of remuneration. The investigation conforms to the principles outlined in the Declaration of Helsinki [15].

### 2.3. Statistics

The mean, standard deviation (SD), median, and interquartile range were used to describe the quantitative data, whereas percentage (%) and frequencies (N) were used for qualitative variables. Reliability coefficients measured by Cronbach’s alpha were calculated for the Hippocratic hypertension self-care scale in order to assess the reproducibility and consistency of the instrument. A Cronbach coefficient alpha value of >0.59 and <0.95 was considered acceptable [15,16]. The underlying dimensions of the scale were checked with an explanatory factor analysis using a Varimax rotation, and the principal components method was used as the usual descriptive method for analyzing grouped data. A factor analysis, using principal component analysis with Varimax rotation, was carried out to determine the dimensional structure of the Hippocratic hypertension self-care scale using the following criteria: (a) eigenvalue > 1; (b) variables should load >0.50 on only one factor and less than 0.40 on other factors; (c) the interpretation of the factor structure should be meaningful; and (d) the scree plot is accurate if the means of commonalities are above 0.60 [16,17]. Bartlett’s test of sphericity with *p* < 0.05 and a Kaiser–Meyer–Olkin (KMO) measure of sampling adequacy of 0.6 were used in carrying out factor analysis. A factor was addressed as significant whether its eigenvalue exceeded 1.0 [16].

A correlation analysis was used to assess internal consistency reliability. The correlation coefficient should not be negative or below 0.20 [18]. Pearson’s rank correlation coefficient was used to check the level of agreement between responses at the test and re-test. Also, a linear regression model with the level of adherence as the dependent variable and one independent variable was used to estimate the correlation between the level of adherence and the added independent variable. The level of significance was 0.05. The analysis was conducted via SPSS 19.0.

## 3. Results

The demographic and clinical characteristics of the sample are presented in Table 1 and Table 2. Almost 55.0% of the sample was women, whereas the mean age was 66.9 years old (range: 30–93 years old). Most participants were divorced or widowed (80.7%), 40.0% had a higher educational level, whereas only 33.2% were employees. More than half the patients had AH stages I or II. The most common self-reported comorbidities were diabetes mellitus (43.4%) and respiratory disease (52.5%).

The median score and the quartiles of all Hippocratic hypertension self-care scale questions are presented in Table 3. The commonalities for the Hippocratic hypertension self-care scale questions are presented in Table 4. The internal consistency characteristics of the Hippocratic hypertension self-care scale showed good reliability, as Cronbach’s alpha was 0.807 for the total scale (items 1–18).

The KMO measure of sampling adequacy was 0.653 and Bartlett’s test of sphericity was 1993.02, df = 153, *p* < 0.001. Factor analysis indicated that there are two principal factors in the model, and these accounted for 92.94%, as presented in Table 5. The first factor (F1) includes the following items: 1 (forget to take medication), 2 (omit to take medication due to its side effects), 3 (omit to take medication when patients feel better), 4 (omit to take medication when patients are outside/travel), and 5 (change the doses according to recommendations); this was termed “Medication aspects”. The second factor (F2) consists of the following items: 6 (daily consumption of fruit and vegetables), 7 (consumption of food responsible for weight increase), 8 (consumption of salty food), 9 (shake salt on your food), 10 (read food labels for ingredients), and 11 (try to lose or maintain body weight); this was termed “Diet aspects”. Cronbach’s alpha was 0.591 and 0.375 for F1 and F2, respectively.

The Hippocratic hypertension self-care scale was well accepted by the participants since it was simple and needed only 10 min to be answered. The items were assessed as relevant, reasonable, unambiguous, and clear. Therefore, face validity was considered very good. According to the test–retest, a high positive correlation was found between the total scores of the assessments (r = 0.995; *p* < 0.001).

The total score on the Hippocratic hypertension self-care scale was significantly lower in patients with less controlled AH (t = 2.168; *p* = 0.036). In addition, the score for the medication sub-scale was significantly higher among participants with less controlled AH (t = 0.744; *p* = 0.012), and the score in the diet subdimension was higher in patients with dyslipidemia (t = 0.658; *p* = 0.013). According to correlation analysis, the level of self-behavior was not associated with age (r = −0.781; *p* > 0.05), gender (t = 0.427; *p* > 0.05), and education level (*p* > 0.05). However, the total score on the Hippocratic hypertension self-care scale was related to the presence of comorbidities and damages in other organs (*p* < 0.01).

## 4. Discussion

The Hippocratic hypertension self-care scale is a non-generic, disease-specific instrument for assessing self-behaviors among patients with AH. Our validation analysis gave a Cronbach’s alpha of 0.807 for the entire scale, whereas the factor analysis detected two main factors; however, further analysis did not show a satisfactory Cronbach’s alpha for these two factors. These domains accounted for 92.94% of the total variance.

To our knowledge, this is the first study to develop a scale assessing all aspects of self-behaviors in patients with AH, which should, therefore, be incorporated into research and clinical practice in order to assess the effectiveness of the provided healthcare and the need for individualized educational intervention. For instance, the Hypertension Self-Care Activity Level Effects (H-SCALE) and Self-Care of Hypertension Inventory (SC-HI) scales assess only the aspects of medication, diet, exercise, body weight, alcohol, and smoking [19,20]. The Hypertension Self-Care Profile (HBP SCP) encompassed the following self-care behaviors: taking medication and lifestyle factors such as exercise, diet, alcohol consumption, non-smoking, self-monitoring of BP, weight control, regular doctor visits, and stress management [21].

The overall Cronbach’s alpha of 0.807 was decoded as high internal consistency for the scale [16,17]. It is essential to mention that the Cronbach’s alpha value was very low for the sub-scales of “Medication”, “Diet”, and “Alcohol”, whereas it could not be calculated for the sub-scales of “Smoking”, “Blood pressure measurement”, and “Exercise” since they included only one item. On the other hand, Cronbach’s alpha was 0.807 for the “Appointment keeping” sub-scale. Therefore, it is clear that the Hippocratic hypertension self-care scale is recommended for use as an entire scale, and each sub-scale is not recommended for use as an independent scale.

The factor analysis of the Hippocratic hypertension self-care scale loaded all items and gave two factors: the “Medication Aspects” (Q1–Q5) and the “Diet Aspects” (Q6–Q11). These two factors account for 92.94% of the total variance. This could be explained by the fact that each sub-section of “Smoking”, “Exercise”, and “Blood pressure measurement” includes only one item, whereas the sub-sections of “Appointment keeping” and “Alcohol consumption” include only two.

Our study provides a significant advantage since the score of the Hippocratic hypertension self-care scale is classified into categories so that healthcare providers can assess the degree to which patients follow the recommended self-behaviors. More specifically, a score over 54 is classified as very good, which means that patients adopt almost all the recommended self-behaviors, a score between 50 and 54 is classified as good, a score between 45 and 50 is classified as fair, and a score below 45 is classified as poor, indicating that patients tend not to follow the recommended self-behaviors.

As for the test–retest, the research team administered the questionnaire two times to the study sample under the same conditions, with an interval of one month. Statistically significant results for the test–retest reliability assessment of the Hippocratic hypertension self-care scale were found during the analysis. More specifically, the correlation coefficient was r = 0.995, which proves the stability of the scale over time (*p* < 0.001).

The results indicated that the total score on the Hippocratic hypertension self-care scale was significantly lower in patients with less controlled AH (t = 2.168; *p* = 0.036). This finding is totally explained since recommendations for lifestyle changes could lead to a significant reduction in BP [22]. More specifically, a low level of score in the subscales of medication and diet is related to a high possibility of uncontrolled AH and dyslipidemia, respectively. Finally, according to correlation analysis, the level of self-behavior was not associated with age (r = −0.781; *p* > 0.05), gender (t = 0.427; *p* > 0.05), and education level (*p* > 0.05). This fact permits the administration of the Hippocratic hypertension self-care scale to the whole population with AH independently of their demographic characteristics.

On the other hand, the total score on the Hippocratic hypertension self-care scale was related to the presence of comorbidities and damages in other organs (*p* < 0.01). Firstly, patients with comorbidities or damage in other organs experience symptoms of many systems and they have to adopt and follow different self-behaviors for each separate health condition. Therefore, the complication of their therapeutic regimen is a burden to them and their level of self-behavior is very low.

The Hippocratic hypertension self-care scale is suggested to be applied in daily clinical practice and may allow healthcare providers to implement specific interventions in order to improve patients’ everyday lives and management of arterial hypertension, rather than focusing solely on the treatment of the specific side effects of the disease.

Our study had some limitations. Firstly, the Hippocratic hypertension self-care scale is a self-administered tool; therefore, information bias could affect the results. Also, we did not conduct ROC analysis due to the lack of a gold-standard tool.

## 5. Conclusions

The Hippocratic hypertension self-care scale showed satisfactory reliability, and the factor analysis indicated two factors that were of interest. We can, therefore, assert that it is a reliable and valid tool for identifying self-behaviors among patients with arterial hypertension. The score of the scale is independent of the demographic characteristics of people with AH; therefore, it could be used for any patient with AH without any limitation. Healthcare providers can use it in their clinical practice to enhance the identification of patients who do not follow and adopt the recommended self-behaviors. Future cross-sectional and cohort studies are suggested so as to inform clinical practicians and guide the development of specific interventions for self-behaviors among patients with arterial hypertension.

## Figures and Tables

**Table 1 healthcare-11-02579-t001:** Demographic characteristics of patients.

Characteristic	N (%)
Gender	
Male	91 (45.0)
Female	111(55.0)
Age (years)	66.9 (11.70)
Education level	
Compulsory	60 (29.7)
Intermediate	60 (29.7)
Secondary/university	82 (40.6)
Marital status	
Married	28 (13.9)
Divorced/widower	163 (80.7)
Unmarried	11 (5.4)
Living conditions	
Alone	17 (8.4)
Family/relation/other support network	185 (91.6)
Employment status	
Employed	67 (33.2)
Unemployed	88(43.6)
Retired	31 (15.3)
Household	16 (7.9)

**Table 2 healthcare-11-02579-t002:** Clinical characteristics and habits of patients.

Characteristic	N (%)
Damage in target organs	
Stroke	15 (6.6)
Stable angina	8 (3.5)
Unstable angina	8 (3.5)
Acute myocardial infarction	4 (1.8)
Retinopathy	9 (3.9)
Comorbidity	
Diabetes mellitus	87 (43.4)
Heart failure	20 (9.9)
Respiratory disease	106 (52.5)
Kidney disease	3 (1.48)
Musculoskeletal disease	35 (17.3)
Classification of hypertension according to ESH	
I	72 (35.6)
II	77 (38.1)
III	29 (14.4)
Isolated systolic hypertension	24 (11.9)
Systolic blood pressure(mmHg)	142 (15.88)
Diastolic blood pressure(mmHg)	86 (10.71)
Blood glucose	106 (36.6)
LDL	119 (41.6)
HDL	46 (12.7)
BMI (kg/m^2^)	22.5 (4.73)
Smoking	
Yes	26 (11.4)
No	176 (87.1)
Daily alcohol consumption	
Yes	26 (11.4)
No	176 (87.1)

ESH: European Society of Hypertension, LDL: low-density lipoprotein, HDL: high-density lipoprotein, BMI: body mass index.

**Table 3 healthcare-11-02579-t003:** Median and quartiles (q25, q75) of the 18 Hippocratic hypertension self-care scale items.

Item	Median	q25	q75
Q1	4.00	3.00	4.00
Q2	3.00	1.00	4.00
Q3	3.00	0.00	4.00
Q4	3.00	0.00	4.00
Q5	1.00	0.00	3.25
Q6	2.00	1.00	4.00
Q7	2.00	1.00	2.25
Q8	2.00	1.75	3.00
Q9	2.00	0.75	4.00
Q10	0.50	0.00	3.00
Q11	2.50	1.00	4.00
Q12	2.00	1.00	3.25
Q13	3.50	1.00	4.00
Q14	3.00	1.00	4.00
Q15	4.00	0.00	4.00
Q16	2.00	1.00	3.00
Q17	4.00	1.50	4.00
Q18	0.00	0.00	0.00

**Table 4 healthcare-11-02579-t004:** Correlation of each item score with their total scores.

	Q1	Q2	Q3	Q4	Q5	Q6	Q7	Q8	Q9	Q10	Q11	Q12	Q13	Q14	Q15	Q16	Q17	Q18
Q1	1.000																	
Q2	0.331	1.000																
Q3	0.191	0.646	1.000															
Q4	0.144	0.634	0.979	1.000														
Q5	0.150	0.383	0.099	0.111	1.000													
Q6	0.202	0.000	0.005	−0.015	0.352	1.000												
Q7	0.129	0.392	0.360	0.315	0.495	0.410	1.000											
Q8	0.292	0.572	0.239	0.223	0.437	0.128	0.531	1.000										
Q9	0.152	0.504	0.475	0.490	0.158	−0.205	0.177	0.475	1.000									
Q10	0.027	0.000	−0.201	−0.204	0.141	0.068	0.276	0.313	0.090	1.000								
Q11	0.141	0.082	−0.061	−0.111	0.181	0.367	0.201	0.318	−0.047	0.092	1.000							
Q12	0.322	0.569	0.491	0.495	0.249	0.122	0.360	0.422	0.579	−0.054	0.343	1.000						
Q13	−0.029	0.679	0.486	0.425	0.062	−0.134	0.111	0.208	0.192	−0.099	−0.016	0.203	1.000					
Q14	0.316	0.583	0.320	0.254	0.178	0.151	0.326	0.416	0.397	0.088	0.056	0.394	0.552	1.000				
Q15	0.182	0.306	0.344	0.250	0.038	−0.001	0.280	0.196	0.106	0.047	0.053	0.357	0.438	0.514	1.000			
Q16	0.297	0.128	−0.200	−0.179	0.315	0.307	0.228	0.471	0.274	0.277	0.470	0.295	−0.145	0.079	−0.117	1.000		
Q17	−0.187	0.270	0.244	0.218	0.042	−0.011	0.062	−0.095	−0.024	0.030	−0.327	−0.261	0.302	0.107	0.164	−0.512	1.000	
Q18	−0.170	0.290	0.294	0.281	0.092	−0.034	0.120	0.014	0.195	−0.156	−0.269	−0.187	0.219	0.090	0.004	−0.354	0.774	1.000

**Table 5 healthcare-11-02579-t005:** Exploratory factors and explained variance after rotation for the Hippocratic hypertension self-care scale.

Factors										Rotation Sums of Squared Loadings		
		Rescaled Loading	Eigenvalues							% of Variance	Cumulative Variance	Cronbach’s Alpha
			Factor 1	Factor 2	Factor 3	Factor 4	Factor 5	Factor 6	Factor 7			
Factor 1	Question 1	0.914	0.917	0.079	0.051	0.220	0.106	0.072	0.052	67.01	67.01	0.591
	Question 2	0.866	0.867	0.166	0.111	0.225	0.115	0.105	0.100			
	Question 3	0.878	0.887	0.018	0.108	0.250	0.087	0.094	0.064			
	Question 4	0.924	0.912	0.066	0.119	0.166	0.188	0.101	0.100			
	Question 5	0.642	0.675	0.146	0.104	0.210	0.102	0.314	0.214			
Factor 2	Question 6	0.668	0.432	0.020	0.544	0.305	0.247	0.173	0.103	25.97	92.94	0.375
	Question 7	0.695	0.238	0.362	0.157	0.162	0.437	0.516	0.416			
	Question 8	0.796	0.093	0.668	0.115	0.134	0.487	0.271	0.171			
	Question 9	0.792	0.169	0.495	0.483	0.218	0.103	0.476	0.276			
	Question 10	0.583	0.093	0.477	0.494	0.085	0.073	0.301	0.100			
	Question 11	0.641	0.232	0.528	0.363	0.304	0.092	0.274	0.074			
Factor 3	Question 12	0.609	0.236	0.524	0.360	0.325	0.017	0.207	0.102			
Factor 4	Question 13	0.733	0.004	0.420	0.435	0.160	0.582	0.056	0.036			0.557
	Question 14	0.599	0.211	0.161	0.303	0.249	0.579	0.199	0.107			
Factor 5	Question 15	0.485	0.078	0.009	0.196	0.110	0.449	0.477	0.208			
Factor 6	Question 16	0.700	0.341	0.599	0.237	0.394	0.110	0.033	0.013			
Factor 7	Question 17	0.851	0.252	0.041	0.497	0.704	0.174	0.113	0.103			0.807
	Question 18	0.826	0.427	0.002	0.368	0.643	0.190	0.241	0.141			

## Data Availability

Not applicable.

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
