# Peer review of "The Development and Validation of the “Hippocratic Hypertension Self-Care Scale”"

_healthcare, 2023, doi:10.3390/healthcare11182579_

Round 1
Reviewer 1 Report
Dear authors,
I read with great interest your manuscript. Indeed, a self-reported scale of hypertension-related patient behavior would be of great relevance since many hypertensive patients fail to adhere to their physicians' recommendations regarding their daily life routine (diet, medication, physical exercise etc.). I think that the Hippocratic scale could be of interest for the clinicians, if its applicability was further tested in other patient cohorts. However, I have some comments that should be addressed:
1. How were the cut-off scores derived? you should explain that in more detail in the Methods section.
2. I would add as a figure the whole scale so that the reader can briefly get its points.
3. How does the scale perform in different subgroup of patients? for example, did patients with less controlled hypertension have lower self-care scores?
4. I would add in the discussion section every relevant self-care scale developed for hypertensive patients in the existing literature to compare the Hippocratic scale with them.
5. You should downgrade the significance of this scale. The population is limited enough and a clinician cannot just integrate the scale in his/her clinical practice unless the value of the scale gets validated in future cohorts.
Be consistent with the abbreviations used and please proofread the manuscript to identify any grammatical or linguistic errors.
Author Response
Reviewer 1
Comment 1: Dear authors,
I read with great interest your manuscript. Indeed, a self-reported scale of hypertension-related patient behavior would be of great relevance since many hypertensive patients fail to adhere to their physicians' recommendations regarding their daily life routine (diet, medication, physical exercise etc.). I think that the Hippocratic scale could be of interest for the clinicians, if its applicability was further tested in other patient cohorts. However, I have some comments that should be addressed.
Response:
Dear Reviewer,
Thank you a lot for your kind comments.
Comment 2: How were the cut-off scores derived? you should explain that in more detail in the Methods section.
Response: We rephrased the paragraph as below: As for the score, we used quartiles to organize data into three points—a lower quartile, median, and upper quartile—to form four groups of the dataset. More specifically, a score over 54 is classified as very good, a score between 50-54 as good, a score between 45-50 as fair, and a score below 45 as poor.
Comment 3: I would add as a figure the whole scale so that the reader can briefly get its points.
Response: We added the total scale in the section of supplementary material.
Comment 4: How does the scale perform in different subgroup of patients? for example, did patients with less controlled hypertension have lower self-care scores?
Response: We conducted more analysis and we added the following paragraph: The total score on the Hippocratic hypertension self-care scale was significantly lower in patients with less controlled AH (t = 2.168; p = 0.036). In addition, the score for the medication sub-scale was significantly higher among participants with less controlled AH (t = 0.744; p = 0.012) and the score in the diet subdimension was higher in patients with dyslipidemia (t = 0.658; p = 0.013). According to correlation analysis, the level of self-behavior was not associated with age (r = -0.781, p > 0.05), gender (t = 0.427, p > 0.05), and education level (p > 0.05). However, the total score on the Hippocratic hypertension self-care scale was related to the presence of comorbidities and damages in other organs (p<0.01).
Comment 5: I would add in the discussion section every relevant self-care scale developed for hypertensive patients in the existing literature to compare the Hippocratic scale with them.
Response: We added the following the paragraph to introduce the relevant self-care scales for hypertension: For instance, the Hypertension Self-Care Activity Level Effects (H-SCALE) and Self-care of Hypertension Inventory (SC-HI) scales assess only the aspects of medication, diet, exercise, body weight, alcohol, and smoking [19-20]. The Hypertension Self-Care Profile (HBP SCP) encompassed the following self-care behaviors: medication taking and lifestyle factors such as exercise, diet, alcohol consumption, non-smoking, self-monitoring of BP, weight control, regular doctor visits, and stress management [21].
Comment 6: You should downgrade the significance of this scale. The population is limited enough and a clinician cannot just integrate the scale in his/her clinical practice unless the value of the scale gets validated in future cohorts.
Response: Dear reviewer, the ratio of question items/patients is 1:10, which is considered appropriate for the validation of a scale. Therefore, the results of the present study are considered adequate. We selected the above ratio since the aim of our study was only to check the reliability and validity of the scale. However, since the prevalence of AH in Greece is about 42% of the general population, researchers who aim to estimate the level of self-behavior in the general population with AH in Greece should include a much bigger sample size. We also added the following sentences in the paper to justify the adequacy of the sample size: The size of the sample was considered appropriate in order for the results of the present study to be considered adequate. Therefore, healthcare providers could be able to use the Hippocratic Hypertension Self-Care Scale without any doubt for the accuracy of their results.
Comment 7: Be consistent with the abbreviations used and please proofread the manuscript to identify any grammatical or linguistic errors.
Response: We checked the whole of the paper for any grammar or spelling issues.
Reviewer 2 Report
I suggest incrementing references to sustain and optimize the discussion and conclusions of the authors.
Please rewrite lines 33 and 34
Please clarify or rewrite lines 48 to 50
Please rewrite lines 55 to 56
Please rewrite or remove frase " On the other hand" in line 59
Please include the questions as supplement material
Please indicate if the number of participants in the trial is significant according to AH prevalence in Greece
Please clarify lines 106 to 110 and separate the inclussion and exclusion criteria
Please check spelling and grammar in all manuscript sections.
Author Response
Reviewer 2
Comment 1: I suggest incrementing references to sustain and optimize the discussion and conclusions of the authors.
Response: Dear reviewer, we incremented the reference list and optimize both the discussion and conclusion section.
Comment 2: Please rewrite lines 33 and 34
Response: We rephrased the lines as below : Arterial Hypertension (AH) is a chronic health condition that may be associated with the development of myocardial disease, stroke and kidney disease.
Comment 3:Please clarify or rewrite lines 48 to 50
Response: We rephrased the sentence as : More specifically, it was found that for people between 40 and 60 years old a reduction by 20 mmHg in systolic BP is associated with a decrease in risk for stroke and mortality of coronary heart disease. Also, it was found the same effect of a reduction of 10 mmHg in diastolic BP.
Comment 4: Please rewrite lines 55 to 56
Response: We rephrased the lines as below: Therefore, we reviewed the existing literature through electronic databases to identify the methods used to assess the level of self-behavior among patients with chronic diseases. In the next stage, we reviewed the available scales assessing self-behavior specific in patients with AH.
Comment 5: Please rewrite or remove frase " On the other hand" in line 59
Response: We replaced the phrase another one as below: At the same time, we identified the Hill-Bone scale which aims to evaluate not only adherence to medication but also adherence to salt consumption and appointment-keeping with healthcare providers
Comment 6: Please include the questions as supplement material
Response: We added the total scale in the section of supplementary material.
Comment 7: Please indicate if the number of participants in the trial is significant according to AH prevalence in Greece.
Response: Dear reviewer, the ratio of question items/patients is 1:10, which is considered appropriate for the validation of a scale. Therefore, the results of the present study are considered adequate. We selected the above ratio since the aim of our study was only to check the reliability and validity of the scale. However, since the prevalence of AH in Greece is about 42% of the general population, researchers who aim to estimate the level of self-behavior in the general population with AH in Greece should include a much bigger sample size. We also added the following sentences in the paper to justify the adequacy of the sample size: The size of the sample was considered appropriate in order for the results of the present study to be considered adequate. Therefore, healthcare providers could be able to use the Hippocratic Hypertension Self-Care Scale without any doubt for the accuracy of their results.
Comment 8: Please clarify lines 106 to 110 and separate the inclussion and exclusion criteria
Response: We separated the inclusion and exclusion criteria as follows:
The study included participants with the below criteria:
1) age over 18 years old,
2) diagnosed AH,
3) prescription of at least one antihypertensive drug,
4) able to read and write Greek,
5) provided written informed consent.
On the other hand, participants with the following criteria were excluded from the study:
1) the presence of a life-threatening disease,
2) the presence of a psychiatric disorder,
3) history of acute myocardial infarction during the last 2 months or cardiac surgery during the last 6 months.
Comment 9: Please check spelling and grammar in all manuscript sections.
Response: We checked the whole of the paper for any grammar or spelling issues.
Round 2
Reviewer 1 Report
Dear authors,
Thank you for improving the manuscript and addressing most of my comments.
However, I cannot still accept the presented results as adequate to be extrapolated to the whole Greek population. These are just unadjusted analyses deriving from a group of people with different comorbidities and we should be mlre cautious when interpreting these results.
Its ok
Author Response
Dear Reviewer,
The guidelines of the American Heart Association mention that patients with hypertension face several common comorbidities that can affect cardiovascular risk and treatment strategies like coronary artery disease, stroke, chronic kidney disease, heart failure, and chronic obstructive pulmonary disease. Like this, the main comorbidities of the sample of the present study were heart failure, pulmonary disease, and diabetes mellitus. Therefore the sample could be representative of the comorbidities based on the literature review. As for Greece, there is observed unfortunately a gap in knowledge regarding the demographic and clinical characteristics of Greek patients with hypertension so it is not possible the compare with other studies.
Reviewer 2 Report
The authors present the manuscript with observations.
I have no comments.
Author Response
Response: Dear Reviewer, thank you a lot for your kind comment.